# Sustainable Omega-3 Lipid Production from Agro-Industrial By-Products Using Thraustochytrids: Enabling Process Development, Optimization, and Scale-Up

**DOI:** 10.3390/foods13223646

**Published:** 2024-11-16

**Authors:** Guilherme Anacleto dos Reis, Brigitte Sthepani Orozco Colonia, Walter Jose Martínez-Burgos, Diego Ocán-Torres, Cristine Rodrigues, Gilberto Vinícius de Melo Pereira, Carlos Ricardo Soccol

**Affiliations:** Department of Bioprocess Engineering and Biotechnology, Polytechnic Center, Federal University of Parana, Rua Cel. Francisco H. dos Santos—100, Curitiba 81530-000, PR, Brazil; guireistda@gmail.com (G.A.d.R.); bricolonia@gmail.com (B.S.O.C.); diego.ocan@ufpr.br (D.O.-T.); cristinelabor@gmail.com (C.R.); gilbertovinicius@ufpr.br (G.V.d.M.P.); soccol@ufpr.br (C.R.S.)

**Keywords:** vegan omega-3 sources, omega-3 fatty acids, DHA (docosahexaenoic acid), thraustochytrids

## Abstract

Thraustochytrids are emerging as a valuable biomass source for high-quality omega-3 polyunsaturated fatty acids (PUFAs), crucial for both human and animal nutrition. This research focuses on cultivating *Schizochytrium limacinum* SR21 using cost-effective agro-industrial by-products, namely sugarcane molasses (SCM), corn steep liquor (CSL), and residual yeast cream (RYC), to optimize biomass and lipid production through a comprehensive multistep bioprocess. The study involved optimization experiments in shake flasks and stirred-tank bioreactors, where we evaluated biomass, lipid content, and DHA yields. Shake flask optimization resulted in significant enhancements in biomass, lipid content, and lipid production by factors of 1.12, 1.72, and 1.92, respectively. In a 10 L stirred-tank bioreactor, biomass surged to 39.29 g/L, lipid concentration increased to 14.98 g/L, and DHA levels reached an impressive 32.83%. The optimal concentrations identified were 66 g/L of SCM, 24.5 g/L of CSL, and 6 g/L of RYC, achieving a desirability index of 0.87, aimed at maximizing biomass and lipid production. This study shows that agro-industrial by-products can be effective and low-cost substrates for producing lipids using thraustochytrids, offering a sustainable option for omega-3 PUFA production. The findings support future improvements in bioprocesses and potential uses of thraustochytrid biomass in food fortification, dietary supplements, nutraceuticals, and as vegan omega-3 sources.

## 1. Introduction

Omega-3 long-chain polyunsaturated fatty acids (ω-3 LC-PUFAs) are valuable for their health benefits. They support brain development, improve bone and eye health, reduce inflammation, and may help prevent hypertension, heart disease, cancer, and Alzheimer’s disease [1]. Historically, fish oil and fish meal have served as the main sources of essential lipids for both human and animal diets. However, rising concerns about contaminants, pollutants, overfishing, and the struggle to satisfy increasing global demand have cast significant doubt on the sustainability of these resources. The swift pace of marine predation has resulted in the depletion of species, jeopardizing biodiversity and creating shortages that the ocean ecosystem cannot replenish rapidly enough [2].

Among all ω-3 LC-PUFAs, α-linolenic acid (ALA) is the most commonly available in human diets, sourced from both plant and animal foods. However, other essential biomolecules, such as eicosapentaenoic acid (EPA) and docosahexaenoic acid (DHA), are more limited, primarily found in fish, fish oils, krill oil, and algae [3]. This trend has spurred a global surge in omega-3 supplementation. Thraustochytrids, marine protists commonly referred to as heterotrophic microalgae, have surfaced as a sustainable and renewable source of valuable lipids. These remarkable organisms can accumulate over 50% of their biomass in the form of intracellular lipids, with more than 25% consisting of omega-3 fatty acids [4]. These organisms are the primary producers of omega-3s in marine ecosystems, with fish acquiring omega-3s by consuming other fish or zooplankton that feed on thraustochytrids [5].

Omega-3-rich thraustochytrid biomass provides a safe and renewable source for human and animal nutrition, suitable for fortified foods, dietary supplements, infant formulas, clinical nutrition, pharmaceuticals, and animal feed [6]. They provide several advantages, such as being non-pathogenic, non-toxic, cost-effective, and sustainable, thriving under controlled fermentation conditions while yielding high levels of lipids. In addition, they are less susceptible to environmental contamination by chemicals and have high lipid yields compared to fish [2,7,8].

The production of thraustochytrids typically involves bioprocessing, which can be costly due to the high expense of the culture medium. However, the agri-food industry generates significant amounts of liquid and solid waste, which, if untreated, can lead to serious environmental, economic, and social issues [7]. Brazil stands as the world’s leading producer of sugar and the second-largest producer of ethanol, creating valuable by-products such as sugarcane molasses and residual yeast cream. Sugarcane molasses (SCM) is a rich, brown, viscous liquid derived from the centrifugation process in sugar manufacturing, containing 54–59.2% sugars and having a pH level ranging from 5.7 to 5.9. For every ton of processed sugarcane, approximately 40 to 60 kg of molasses are produced. While its primary use lies in ethanol production, SCM is also instrumental in various sectors, including biotechnology, pharmaceuticals, chemicals, and food industries [9].

Residual yeast cream (RYC) is an agro-industrial by-product produced during the biotechnological processes of industrial ethanol and beer production. Following alcoholic fermentation, approximately 75–85% of the yeast cells remain viable, offering opportunities for reuse in subsequent fermentations. The leftover 20% becomes residual yeast cream, which boasts a protein content ranging from 33% to 50%. RYC undergoes washing, desalinization, and spray-drying, with cane yeast incorporated into animal feed and brewer’s yeast utilized as a nutritional supplement in human food [9,10].

Corn and its by-products yield significant quantities of corn steep liquor (CSL), particularly in countries like Brazil and the United States. CSL is characterized by its light to dark brown color, ensiled aroma, and low pH level (between 3.7 and 3.86) due to its high lactic acid content, alongside an impressive protein concentration ranging from 31.6% to 49.0%. This valuable substance is produced through the wet milling process in corn processing, where corn kernels are soaked in heated water and then successively macerated. Recently, CSL has found its way into animal feed and is increasingly being used as a substitute for yeast extract in fermentation processes [11].

This study aimed to produce lipid-rich thraustochytrids using low-cost agro-industrial by-products. Corn steep liquor (CSL) and residual yeast cream (RYC) were used as nitrogen sources for biomass growth, while sugarcane molasses (SCM) provided carbon for lipid accumulation. The research involved improving strains and optimizing cultivation through precision fermentation and design of experiments (DOE) to enhance biomass, lipid, and DHA production. The goal was to scale up the process in a 10 L stirred-tank bioreactor to produce polyunsaturated fatty acids (PUFAs) for food fortification and human nutrition.

## 2. Materials and Methods

### 2.1. Strain Preparation

The thraustochytrid organism that was utilized for this research was *Schizochytrium limacinum* SR21 (ATCC^®^ MYA-1381™), which was preserved at −80 °C in 20% glycerol. The strain was reactivated in GYPS-MSG broth (15 g/L glucose, 2 g/L yeast extract, 1 g/L peptone, 10 g/L sea salts, and 2 g/L monosodium glutamate) and incubated in a shaker at 28 °C for 2 days. This reactivation methodology was obtained from Colonia’s thesis, 2021 [12].

### 2.2. Bioprocess Optimization in Shake Flasks

The enhancement of biomass and lipid production through the use of SCM, CSL, and RYC was executed using a Central Composite Design (CCD). This approach was built upon the experimental framework established in Colonia’s doctoral thesis from 2021 [12].

The Central Composite Design (CCD 23, 4 center points, −1.68, −1.0, 0, +1.0, +1.68 levels) was used to optimize the production of biomass and lipids. The materials used were sugarcane molasses (SCM: 66, 80, 100, 120, and 134 g/L), corn steep liquor (CSL: 2, 8, 17, 26, and 32 g/L), and residual yeast cream (RYC: 2.6, 4, 6, 8, and 9.4 g/L) (Table 1).

The following fixed parameters were established for the cultivation process: an inoculum rate of 10%, a culture medium pH of 5.5, a sea salt concentration of 10 g/L, an agitation speed of 120 rpm, and a temperature set at 28 °C. The cultivation occurred in 250 mL flasks, each containing 50 mL of culture medium, and was incubated on a shaker for a duration of 5 days.

The data was analyzed in the software STATISTICA 7 (StatSoft, Hamburg, Germany) using the Pareto charts from ANOVA, observed values vs. predictive values, and response surface methodology (RSM) with a significance of 5%. These studies generated mathematical models for desirability profiling and optimization according to Equation (1).
Y = β_o_ + β_1_X_1_ + β_2_X^2^ + β_3_X_2_ + β_4_X^2^ + β_5_X_3_ + β_6_X^2^(1)

### 2.3. Bench-Scale Strategies

A pre-inoculum was created in a 250 mL flask with 70 mL of culture broth, formulated with 50 g/L glucose, 5 g/L yeast extract, 2 g/L peptone, and 15 g/L sea salts, adjusted to a pH of 6.0. This flask was incubated in a shaker at 120 rpm and 25 °C for 24 h. Following this, the inoculum was prepared in a 2 L Erlenmeyer flask containing 630 mL of culture medium, which included 50 g/L SCM, 10 g/L CSL, 2 g/L RYC, and 10 g/L sea salts, also with a pH of 6.0. To this 2 L flask, 70 mL of the pre-inoculum (10%) was added, and the mixture was incubated at 120 rpm and 28 °C for 48 h, promoting optimal growth.

The submerged batch strategies were developed on a bench scale in a 10 L stirred-tank bioreactor (STBR, New Brunswick Scientific BioFlo^®^ 115 Fermentor, Eppendorf, Hamburg, Germany) with 7 L working volume (30% headspace).

The batch media was 6.3 L containing 90 g/L SCM, 25 g/L CSL, 15 g/L RYC, 10 g/L sea salts, and 2 g/L MSG. The initial pH was adjusted to 5.5–6.0 with KOH 5 M. The bioreactor vessel was sterilized at 121 °C and 1 atm for 30 min.

The inoculum rate was 10% (700 mL). Then, 45 g/L sugarcane molasses was fed to the bioreactor when the concentration of reducing sugars dropped to less than 20 g/L. The bioprocess was controlled at 28 °C, the aeration rate at 0.5–1.0 vvm, and the agitation speed at 200–600 rpm to maintain a 30% dissolved oxygen (DO) for 120 h. The addition of Nitrofoam anti-foaming H10 effectively mitigated foam generation.

### 2.4. Kinetic Studies of Bioprocess

Every 12 h, 15 mL samples of the culture broth were extracted from the bioreactor and subjected to centrifugation at 5000 rpm for 10 min. The biomass, represented as dry cell weight, was measured by drying the samples in an oven at 105 °C until they reached a constant weight.

The consumption of reducing sugars (substrate utilization) was assessed using the previously adapted DNS method (3,5-dinitrosalicylic acid) [13]. Then, 100 µL of the supernatant was diluted in 900 µL of distilled water. The assay was performed in 2 mL microtubes, where 25 µL of the diluted supernatant and 25 µL of the DNS reagent were added. The microtubes were boiled at 100 °C for 5 min. Then, the microtubes were cooled in an ice bath and 330 µL of distilled water was added. Aliquots of 300 µL were transferred in a 96-well transparent microplate with a flat bottom, and absorbance was measured at 540 nm in a microplate spectrophotometer reader (Synergy™ HTX, BioTek Instruments, Winooski, VT, USA).

The results were plotted for studies of batch growth kinetics as described in Equations (2)–(17) in Table 2.

### 2.5. Biomass Recovery and Proximate Composition

The total biomass from the bioreactors was harvested through centrifugation, then transferred to trays and dried in a circulating air oven at 45 °C for 36 h. After cooling to room temperature, the dry biomass was processed in a knife mill and sifted through a 35-mesh screen.

#### Lipid Content

The lipid content was determined using an adapted cold extraction method [14,15]. An analytical balance was used to weigh 0.2 g of dry biomass. Following this, 0.8 mL of deionized water was introduced, and the mixture was vortexed for 10 s. Next, 1 mL of chloroform and 2 mL of methanol were added, followed by stirring for 2 min. To ensure proper phase separation and achieve a ratio of 2:2:1.8 of chloroform, methanol, and water, an additional 1 mL of deionized water was incorporated and stirred for 30 s. The samples were then centrifuged at 5000 rpm for 6 min. After centrifugation, a 1.4 mL aliquot of the lower phase was carefully transferred to pre-dried and pre-weighed Petri dishes. The chloroform in the dishes was subsequently evaporated at 45 °C for 3 h in a vacuum oven. The lipid content was calculated according to Equation (18).
(18)$$Lipid\;content\;\left(\%\right)=\frac{Weight\;of\;lipid\;in\;aliquot\;\left(g\right)\;*\;\frac{Volume\;of\;chloroform\;layer\;\left(ml\right)}{Volume\;of\;aliquot\;\left(ml\right)}}{Weight\;of\;dry\;biomassa\;sample\;\left(g\right)}\;*\;100$$

### 2.6. Fatty Acids Analysis

The fatty acid methyl esters (FAMEs) were obtained after the direct transesterification of biomass according to previous work [16]. Fatty acid methyl esters (FAMEs) were produced through the direct transesterification of freeze-dried cell biomass using a modified protocol. Approximately 25–40 mg of freeze-dried biomass was placed in 16 mL glass tubes sealed with PTFE/Teflon caps. To this, 2.0 mL of 15% H_2_SO_4_ in methanol was introduced and the mixture was vortexed for 30 s. Following this, 2.0 mL of chloroform was added, and the solution was vortexed for an additional 10 s. The tubes were then submerged in a water bath set to 90 °C for 40 min, with manual shaking using tweezers every five minutes. Once finished, the tubes were allowed to cool to room temperature.

The reaction mixtures were transferred to 15 mL centrifuge tubes, and 1 mL of milli-Q water was added. These tubes were vortexed for 45 s, followed by centrifugation at 5000 rpm for five minutes. The lower phases were then dried in clean tubes containing anhydrous sodium sulfate, vortexed for 10 s, and allowed to rest for 15 min. The organic phases were meticulously filtered through absorbent cotton to eliminate any potential impurities. The fatty acid methyl esters (FAMEs) were subsequently stored in chromatography vials, which were sealed and kept at −20 °C until GC analysis.

The fatty acid composition was analyzed by gas chromatograph (GC-2010 Plus Capillary GC, Shimadzu Scientific Instruments, Columbia, MD, USA) and compared to the Supelco^®^ 37 Component FAME Mix standard (Sigma-Aldrich, Merck, Darmstadt, Germany). The injection volume was set at 1 µL with a split ratio of 1:10. The carrier gas, helium, was maintained at a linear speed of 32.5 cm/s. The injector and FID detector temperatures were meticulously controlled at 240 °C and 250 °C, respectively. The oven column temperature commenced at 100 °C for an initial 5 min, followed by an increase to 240 °C at a rate of 4 °C/min, where it was held for an additional 5 min.

## 3. Results and Discussion

### 3.1. Bioprocess Modeling and Optimization

The key variables—including SCM, CSL, and RYC—identified during the optimization phases I and II of Colonia’s 2021 study [12] were analyzed using the central composite design (CCD 23) to enhance biomass and lipid production (refer to Table 3). *S. limacinum* SR21 exhibited outstanding performance, yielding 17.60 g/L of biomass, a lipid content of 29.14%, and lipid production of 4.49 g/L.

Researchers successfully achieved a biomass concentration of 25 g/L and a lipid content of 25.5%, translating to 6.38 g/L of lipids after optimization using a Central Composite Design (CCD). This was accomplished with 70 g/L of soybean cake meal (SCM), 30 g/L of corn steep liquor (CSL), and 30 g/L of sodium glutamate, grown in a 500 mL shake flask under conditions of 165 rpm. The temperature was reduced from 28 to 20 °C over a period of four days using the mutant strain *Schizochytrium* sp. SHG104 [17].

A recent study focused on optimizing the production of biomass, lipids, and DHA using the strain *Aurantiochytrium* sp. R2A35. Researchers investigated the effects of varying the volume ratio of sugar cane molasses, the carbon/nitrogen ratio, and citric acid concentration. The ideal parameters identified were 25% (*v*/*v*) sugar cane molasses, a carbon-nitrogen ratio of 60, and 0.2 g/L of citric acid. Under these optimized conditions, the production levels achieved were 34.88 g/L of biomass, 15.17 g/L of lipids, and 6.09 g/L of DHA. Additionally, by adjusting the aeration conditions, DHA production further increased to 10.04 g/L. The findings indicated that high oxygen levels enhanced biomass accumulation, while a lower oxygen supply led to greater DHA accumulation in *Aurantiochytrium* sp. R2A35 [18].

Moreover, Pareto graphs illustrate that enhancing CSL and RYC effectively stimulates biomass production with notable linear effects (see Figure 1). One reason the SCM does not play a more impactful role in thraustochytrid biomass is that it achieves optimal concentrations, unlike lipid production, where a significant linear decrease in SCM coincides with a substantial positive quadratic effect. This phenomenon indicates that while elevated SCM levels may inhibit cell growth due to substrate overload, they remain crucial for promoting lipid accumulation within the cells.

Conversely, the linear effect of RYC on lipid accumulation was found to be insignificant, while the reduction in CSL significantly influenced lipid accumulation in a quadratic manner. Additionally, the CCD facilitated the development of mathematical regression models that accurately represented the response variables. Four models were assessed based on R_2_, adjusted R_2_, and F-test results for lack of fit. The linear/quadratic model encompassing the main effects was chosen for all three response variables (Y1, Y2, and Y3), formulated in Equations (19), (20), and (21). Here, Y1 represents biomass (g/L), Y2 denotes lipid content (%), Y3 indicates lipids (g/L), and X1, X2, and X3 correspond to sugarcane molasses (g/L), corn steep liquor (g/L), and residual yeast cream (g/L), respectively.
Y1 = −5.5503 + 0.0296X1 − 0.0002X1^2^ + 1.0278X2 − 0.0180X2^2^ + 1.8458X3 − 0.1132X3^2^(19)
Y2 = 38.9751 − 0.7463X1 + 0.0032X1^2^ + 1.4421X2 − 0.0404X2^2^ + 3.6205X3 − 0.3326X3^2^(20)
Y3 = 4.6544 − 0.1370X1 + 0.0006X1^2^ + 0.3412X2 − 0.0083X2^2^ + 0.7763X3 − 0.0654X3^2^(21)

The predictive values generated by these models were meticulously compared to the observed data, as illustrated in Figure 1. The biomass values closely aligned with the trend line and remained within the regression limits, with only a handful of points deviating from the confidence interval. This alignment is indicative of a higher R^2^ value in biomass production, which reflects a superior fit to the experimental data. Conversely, the lipid values exhibited greater dispersion, with several points lying outside the regression area, highlighting a higher variability in the lipid production data.

The R-squared values for lipids were lower; however, the predictors continued to show statistical significance. This phenomenon may be explained by various factors, including variability in cell replication, cellular stress, or changes in lipid metabolism, all of which could have influenced the results.

### 3.2. Prediction and Desirability Profiling

The desirability index (DI) was assessed by examining various combinations for multicriteria optimization. It became evident that when focusing solely on the biomass variable (Y1) in the optimization process, high levels of SCM (X1), CSL (X2), and RYC (X3) would yield the maximum production according to the predictive model, achieving a DI of 1.0. However, this approach overlooks lipid production (Y2), which, when evaluated independently, indicates that lower values of SCM, SCL, and RYC are more effective for enhancing lipid accumulation, resulting in a DI of 0.87.

Conversely, a desirability index of 0.87 was observed when applying the criteria Y1 and Y2. This outcome aligns with the multicriteria analysis incorporating Y1, Y2, and Y3, utilizing identical predictive values. Thus, the multicriteria approach involving Y1, Y2, and Y3 proved to be optimal for evaluating the three response variables: biomass (g/L), lipid content (%), and lipids (g/L), achieving an overall desirability index of 0.87 (as shown in Figure 2).

The best media formula for achieving multiple outcomes includes 66 g/L SCM, 24.5 g/L CSL, and 6 g/L RYC. This combination predicts a biomass yield of 17.00 g/L (range: 14.24 to 19.75 g/L) and a lipid content of 24.49% (range: 21.06 to 27.92%). The expected lipid yield is 4.00 g/L (range: 3.39 to 4.57 g/L). To reach a desirability index (DI) of 1.0, the process must produce 17.60 g/L of biomass, 29.14% lipid content, and 4.49 g/L of lipid. Response surfaces from regression models highlight optimized regions (see Figure 3). The tested SCM values spanned the optimal biomass production area, making SCM not a significant variable in the Pareto chart (Figure 1), as it was already within the optimal range.

Biomass data closely adhered to the trend line and remained within the regression limits, with only a few values falling outside the confidence area. This consistency can be attributed to a higher R^2^ value for biomass production, indicating a stronger correlation with the obtained data. In contrast, lipid data exhibited a wider dispersion, with some points straying beyond the regression area. Although the R^2^ for lipids was lower, the predictors were still statistically significant. Potential explanations for this variability include errors during the optimization and experimentation processes, fluctuations in cell duplication, cell stress, or shifts in lipid metabolism [12].

The SCM set at 66 g/L facilitated a more optimized response surface for biomass production when combined with the CSL and RYC variables, both within their optimal ranges. A similar trend was observed in the surfaces generated for lipid production, which displayed clearly defined optimized areas. Nonetheless, these surfaces indicated a tendency for lipid production to rise as SCM levels decreased, highlighting the onset of the optimal region.

Optimizing carbon/nitrogen sources, process variables, and culture medium significantly boosted lipid production in *S. limacinum* SR21 without genetic modifications. The strain showed promising yields in a bioprocess using agro-industrial by-products, making it suitable for scaling up.

### 3.3. Batch Bioreactor Studies

The *S. limacium* SR21 strain was chosen for scale-up in a 10 L bioreactor, along with fermentation kinetic studies. Table 4 presents a comparative analysis highlighting the differences in fermentation conditions between the Erlenmeyer flasks and the bioreactor. To achieve approximately 30% dissolved oxygen, we optimized shaker medium concentrations by adjusting the impeller agitation speed and aeration through a series of 10 L bioreactor fermentations.

The SCM concentration was elevated to 90 g/L, taking advantage of the batch mode operation that entails no additional feed during fermentation. This adjustment facilitated a thorough assessment of substrate consumption and residual reducing sugars utilized by the cells, resulting in increased weight gain driven by fat accumulation linked to nitrogen limitation. The CSL concentration closely matched the optimized shaker condition of 24.5 g/L. Furthermore, the concentration of RYC was increased to 15 g/L to enhance cell growth and biomass production in the initial phases of fermentation. Nitrogen sources play a crucial role in the metabolic efficiency of proteins that drive the proliferation of microorganisms during this process [19]. Additionally, 2 g/L of monosodium glutamate (MSG) was added to the bioreactor to boost fatty acid production.

After 120 h in the batch bioreactor, biomass production reached 39.29 g/L, lipid content was 38.13%, DHA made up 32.83% of total lipids, and lipid production totaled 14.98 g/L (Figure 4A,B). These results represent increases of 2.27-fold, 1.50-fold, and 3.40-fold compared to cultivation in 250 mL flasks. Additionally, 90% of the fatty acids produced were pentadecanoic (C15:0), hexadecanoic (C16:0), heptadecanoic (C17:0), and docosahexaenoic acids. Figure 5 provides a detailed illustration of the spectrum, highlighting the identification of compounds associated with their corresponding peaks.

Research has explored various strategies to enhance NADPH levels in culture media, including the incorporation of MSG and malic acid. MSG serves as a nitrogen source that can effectively mimic aerobic glycolysis by boosting the activity of glucose-6-phosphate dehydrogenase (G6PDH). This enzyme is crucial as it increases NADPH production, which is vital for the pentose phosphate pathway in the cytosol of non-photosynthetic cells. Furthermore, NADPH plays a critical role as a reducing agent in the enzyme complex responsible for the elongation and formation of double bonds, essential for the synthesis of polyunsaturated fatty acids (PUFAs) [2].

A study demonstrated that the addition of MSG to a medium with treated cane molasses significantly enhanced both biomass and DHA content produced by *Schizochytrium* sp. CCTCC M209059. The biomass increased from 29.65 g/L to 36.6 g/L, while the DHA content rose from 38.71% to 42.39% [16]. Another study containing 80 g/L glucose, 10 g/L yeast extract, 10 g/L MSG, 15 g/L sea salt, 2 g/L MgSO4·7H_2_O, and 1 g/L KH_2_PO_4_ in the cultivation of *Schizochytrium* sp. ATCC 20888 obtained 55.83 g/L of biomass, 34.97% lipid content, 19.50 g/L of lipids, and 35.68% of DHA [20].

By varying MSG concentrations, researchers found that high levels reduce lipid content, while low levels increase DHA content. *Aurantiochytrium* sp. SW1 produced 24.46 g/L biomass with 38.46% lipid content (9.40 g/L lipids) when grown in an optimized medium containing 60 g/L glucose, 2 g/L yeast extract, 24 g/L monosodium glutamate, and 6 g/L sea salt in a 5 L bioreactor at 30 °C, with 1 vvm aeration and 200 rpm impeller speed for 96 h [21].

The logarithmic cell growth curves observed in 250 mL shake flasks and a 10 L bioreactor exhibited comparable profiles (Figure 4C). Notably, the maximum specific growth rate (µ) increased from 0.59 day-1 in the shaker to 0.66 day^−1^ in the bioreactor, leading to a reduction in cell doubling time from 28.41 to 25.11 h (Table 5). Additionally, the maximum yields improved, with biomass reaching 0.64 g/L.h (rX max) and lipid production at 0.35 g/L.h (rP max).

The substrate consumption productivity was measured at 0.74 g/L.h (rS max). Starting with an initial solid carbon material (SCM) concentration of 90 g/L, which translated to 60.25 g/L of reducing sugars, we observed a decrease to 8.25 g/L of reducing sugars by the end of fermentation. Consequently, the maximum specific substrate consumption rates (QS max) achieved were 0.12 h^−1^ for shake flask cultivation and 0.09 h^−1^ for bioreactor cultivation at the 24-h mark, as illustrated in Figure 4D.

In the bioreactor, the biomass-substrate yield (YX/S) was measured at 0.64 g biomass/g substrate, while the product-substrate yield (YP/S) stood at 0.29 g lipids/g substrate. These results closely align with the values presented in the graphical data and equation (Figure 4E,F). The notable enhancement in both YX/S and YP/S yields in the bioreactor, when compared to the cultures grown in shaken flasks, indicates a more effective utilization of the nutrients within the culture medium by the wild-type strain. This improvement highlights the strain’s superior efficiency in assimilating these nutrients into biomass and products through cellular metabolism. Furthermore, the product-biomass yield (YP/X) has significantly increased from 0.29 g lipids/g biomass in the shaker setups to 0.45 g lipids/g biomass in the bioreactor.

*Aurantiochytrium* sp. SW1 was recently assessed in a medium enriched with agroindustrial by-products, specifically 100 g/L of MD-2 pineapple extract, 14 g/L of MSG, 50% salinity, and 6 g/L of yeast extract. This evaluation took place in a 5 L batch bioreactor at 28 °C, with a stirring speed of 500 rpm and an airflow of 1 vvm over a duration of 120 h. Remarkably, the results revealed peak values of biomass at 31.87 g/L, lipids at 18.85 g/L, and DHA at 8.20 g/L, showcasing the potential of MD-2 extract as an effective alternative to commercial glucose and fructose sources [22]. Factors such as aeration and agitation speed play a crucial role in significantly enhancing process yields in various biotechnological applications. By optimizing these parameters, we can achieve better homogenization of nutrients within the growth medium, which is essential for cell development. Additionally, these factors help to increase the concentration of dissolved oxygen, facilitating aerobic processes. Furthermore, they also improve mass and heat transfer, as well as enhance the suspension of cells in the liquid medium, contributing to a more efficient and productive environment for microbial growth [23].

*A. limacinum* BUCHAXM 122 produced 43.05 g/L of biomass and a DHA yield of 0.142 g/g CDW in GYP medium (60 g/L glucose, 10 g/L yeast extract, 10 g/L peptone) after 48 h in a 2 L STBR with a 5% inoculum rate, 600 rpm, 25 °C, and 2 vvm. However, by the end of fermentation at 120 h, biomass concentration and DHA yield decreased to 30.65 g/L and 0.113 g/g CDW, respectively [24]. High aeration (2 vvm) caused foaming, while high agitation speed (600 rpm) reduced DHA yield due to cell damage from mechanical shear stress. In the STBR experiments, we also observed foaming with aeration between 0.5 and 1 vvm and agitation speeds of 400 to 800 rpm, which we managed using antifoam.

A recent comparative study utilizing *Schizochytrium* sp. and the same optimization techniques as this research, specifically through Central Composite Design, yielded comparable results. The experiments revealed that the optimal conditions for cultivation were a duration of 120 h, with a nutrient medium consisting of 118.71 g/L of glucose, 20.00 g/L of sodium glutamate, and 15.16 g/L of sea salt. Ultimately, the biomass produced was 39.23 ± 0.56 g/L, along with DHA and lipid yields of 8.33 ± 0.074 g/L and 30.24 ± 2.66 g/L, respectively [25]. This signifies that the current research is aligning with ongoing trends in optimization, even when employing different substrates.

## 4. Conclusions

The enhancement of thraustochytrids through various strategies, including the optimization of bioprocesses via Design of Experiments (DOE) methodology, has led to the remarkable production of lipid-rich biomass. By employing bioprocess modeling, we developed predictive profiles based on desirability, enabling us to identify optimal conditions for scale-up in stirred-tank bioreactors using a wild-type strain. The results were impressive, with maximum yields reaching 39.29 g/L (0.64 g/L.h) in biomass, 14.98 g/L (0.35 g/L.h) in lipids, 32.83% DHA concentration, and a lipid-biomass yield (YP/X) of 0.45 g/g CDW. Moreover, agro-industrial by-products such as Substrate Concentrated Medium (SCM), Corn Steep Liquor (CSL), and Residual Yeast Concentrate (RYC) emerged as promising alternatives to conventional synthetic and defined sources. Future enhancements to this process could be achieved by optimizing bioreactor conditions, including the operating mode (batch, fed-batch, continuous), feeding strategies, aeration, impeller agitation speed, pH and foam control, lipid accumulation regulation, and temperature adjustments, among other factors. Once optimal production conditions are established, research can commence on the most effective methods for deodorization and the incorporation of PUFAs in commonly consumed foods such as meat, eggs, dairy products, and infant formulas.

## Figures and Tables

**Figure 1 foods-13-03646-f001:**
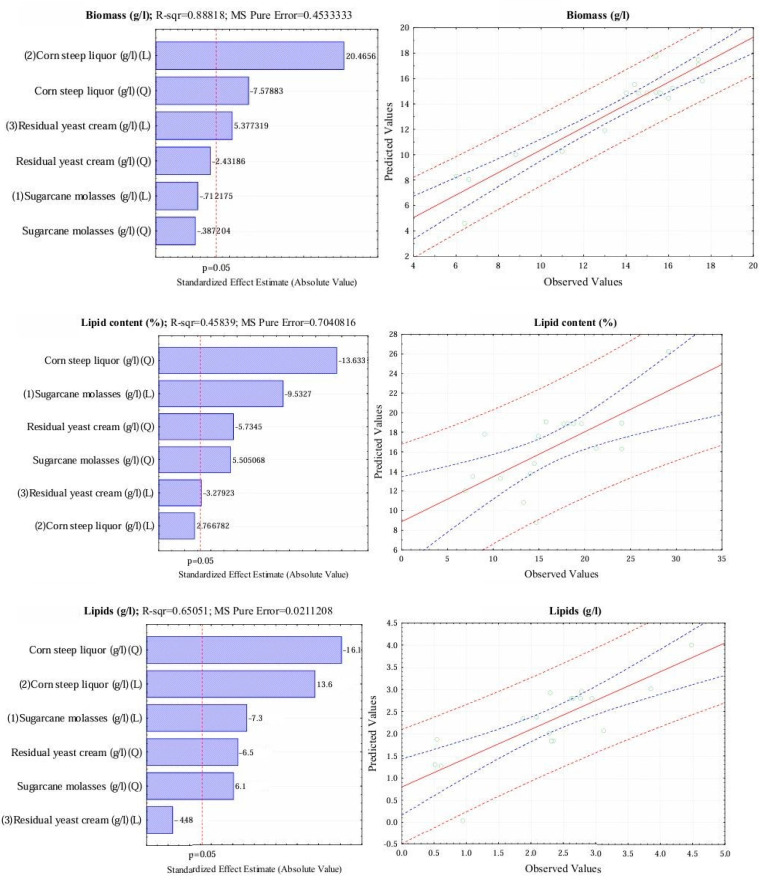
The Pareto chart shows standardized effects and compares predicted vs. observed values in the central composite design (CCD) for biomass and lipid production, using a 5% confidence level (blue dotted line) and prediction level (red dotted line).

**Figure 2 foods-13-03646-f002:**
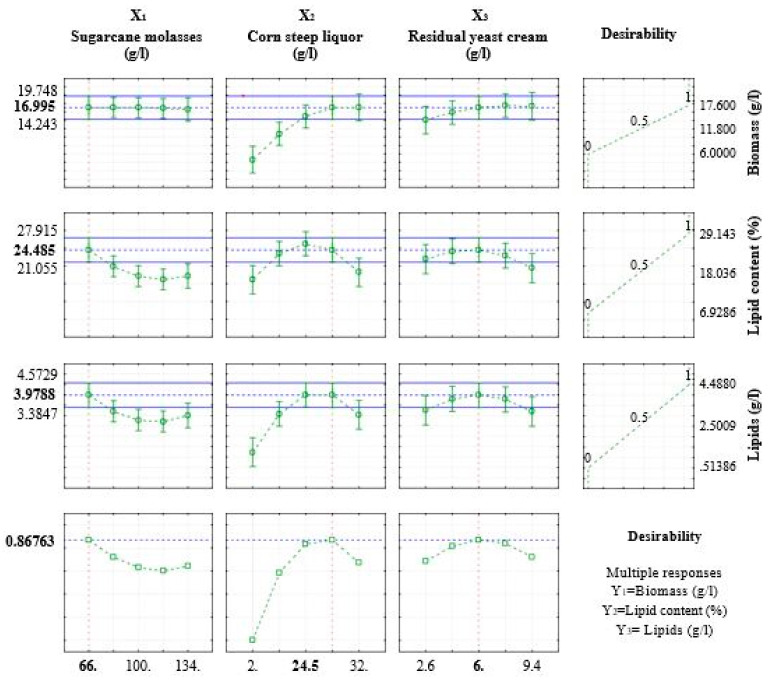
Prediction profiler and desirability for optimization of multiple responses Y1: Biomass (g/L), Y2: Lipid content (%), and Y3: Lipids (g/L).

**Figure 3 foods-13-03646-f003:**
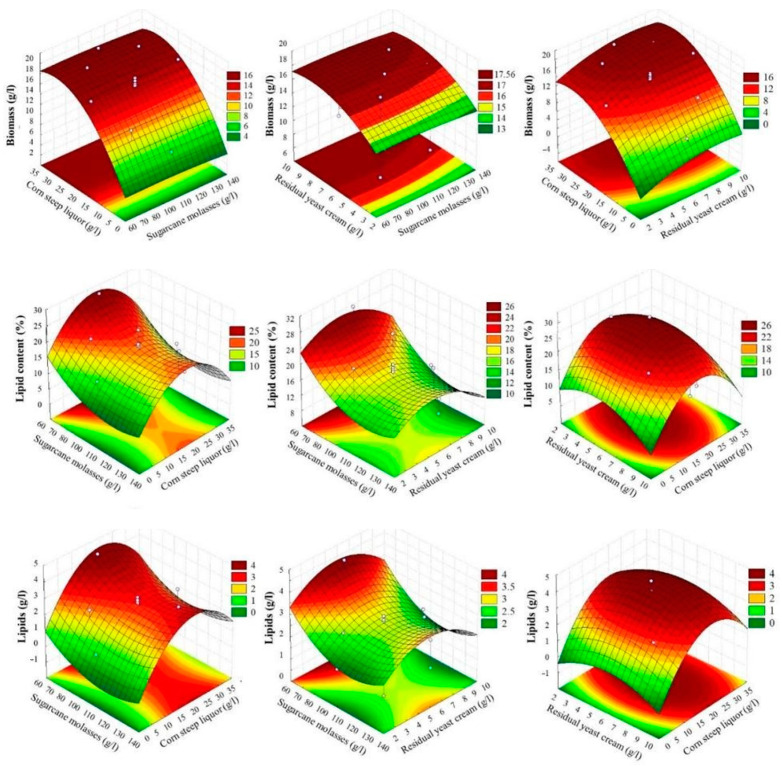
Response surfaces from CCD 23 regression models optimize bioprocess using sugarcane molasses (66 g/L), corn steep liquor (24.5 g/L), and residual yeast cream (6 g/L) based on predictions and desirability profiles.

**Figure 4 foods-13-03646-f004:**
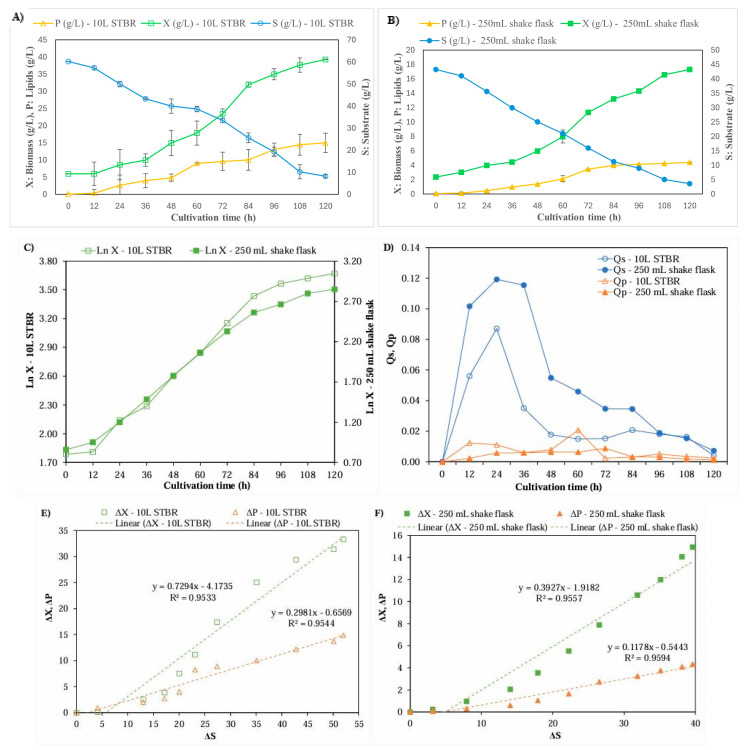
Batch kinetic curves of a bioprocess using thraustochytrid and agro-industrial by-products as substrate include: (**A**) Scale-up in a stirred-tank bioreactor (STBR). (**B**) 250 mL shake flask. (**C**) Logarithmic cell growth (Ln X). (**D**) Substrate consumption rate (Qs) and product formation rate (Qp). (**E**) Yield evolution curves for changes in biomass (ΔX), product (ΔP), and substrate (ΔS) in a 10 L STBR. (**F**) Yield evolution curves for changes in the same parameters in a 250 mL shake flask. The standard deviations depicted in Figure 4B were so minute that they were undetectable within the confines of the image.

**Figure 5 foods-13-03646-f005:**
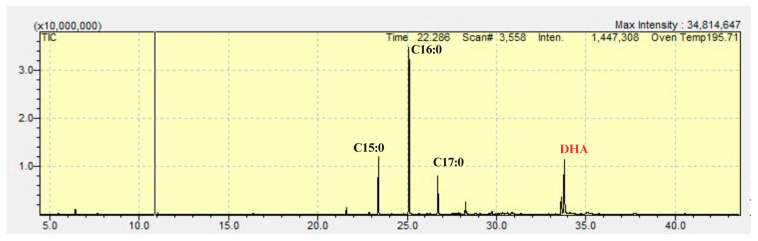
The objective of this study is to employ gas chromatography spectroscopy to identify the specific fatty acids present in the biomass. As can be observed, the four most prominent peaks represent pentadecanoic (11.54%), hexadecanoic (34.28%), heptadecanoic (7.34%), and docosahexaenoic (36.65%) fatty acids.

**Table 1 foods-13-03646-t001:** Central Composite Design (CCD 23) for the optimization of biomass and lipid.

Independent Variables	−1.68	−1	0	+1	+1.68
Sugarcane molasses (g/L)	66	80	100	120	134
Corn steep liquor (g/L)	2	8	17	26	32
Residual yeast cream (g/L)	2.6	4	6	8	9.4

**Table 2 foods-13-03646-t002:** Equations for kinetic determinations in batch bioreactor.

Kinetic Determinations	Equation
Maximum specific growth rate (h^−1^)	$\mu \max=\frac{lnX2-lnX1}{t2-t1}$ (2)
Doubling time of cells (h)	$\tau d=\frac{ln2}{\mu \;\max}$ (3)
Maximum biomass productivity (g∙L^−1^∙h^−1^)	$rx\max=\frac{\Delta X}{\Delta t}=\frac{X2-X1}{t2-t1}$ (4)
Total biomass productivity (g∙L^−1^∙h^−1^)	$rx\;total=\frac{Xf-XI}{tf-ti}$ (5)
Overall biomass productivity (g∙h^−1^)	rx overall=rx total∗V (6)
Maximum substrate consumption productivity(g∙L^−1^∙h^−1^)	$rs\max=\frac{\Delta S}{\Delta t}=\frac{S1-S2}{t2-t1}$ (7)
Total substrate consumption productivity (g∙L^−1^∙h^−1^)	$rs\;total=\frac{Si-Sf}{tf-ti}$ (8)
Overall substrate consumption productivity (g∙h^−1^)	rs overall=rs total∗V (9)
Maximum product formation productivity (g∙L^−1^∙h^−1^)	$rp\max=\frac{\Delta P}{\Delta t}=\frac{P2-P1}{t2-t1}$ (10)
Total product formation productivity (g∙L^−1^∙h^−1^)	$rp\;total=\frac{\Delta P}{\Delta t}=\frac{Pf-Pi}{tf-ti}$ (11)
Overall product formation productivity (g∙h^−1^)	rp overall=rp total∗V (12)
Specific substrate consumption rate (h^−1^)	$Qs\max=rs\max*\;\frac{1}{X}$ (13)
Specific product formation rate (h^−1^)	$Qp\max=rp\max*\;\frac{1}{X}$ (14)
Biomass-substrate yield (gbiomass∙gsubstrate^−1^)	Y_X/S_ = $\frac{\Delta X}{\Delta S}=\frac{Xf-Xi}{Si-Sf}$ (15)
Product-substrate yield (gproduct∙gsubstrate^−1^)	Y_P/S_ $=\frac{\Delta P}{\Delta S}=\frac{Pf-Pi}{Si-Sf}\;$ (16)
Product-biomass yield (gproduct∙gbiomass^−1^)	Y_P/X_ $=\frac{\Delta P}{\Delta X}=\frac{Pf-Pi}{Xf-Xi}$ (17)

**Table 3 foods-13-03646-t003:** Central Composite Design and four center points (C) for the production of lipid-rich thraustochytrid biomass.

Run	Independent Variables			Response Variables		
	Sugarcane Molasses (g/L)	Corn Steep Liquor (g/L)	Residual Yeast Cream (g/L)	Biomass (g/L)	Lipid Content (%)	Lipids (g/L)
1	80.0	8.0	4.0	6.00	9.07%	0.54
2	80.0	8.0	8.0	11.00	21.29%	2.34
3	80.0	26.0	4.0	17.60	15.79%	2.78
4	80.0	26.0	8.0	15.40	14.93%	2.30
5	120.0	8.0	4.0	6.60	7.79%	0.51
6	120.0	8.0	8.0	8.80	6.93%	0.61
7	120.0	26.0	4.0	14.40	14.50%	2.09
8	120.0	26.0	8.0	17.40	10.79%	1.88
9	66.0	17.0	6.0	15.40	29.14%	4.49
10	134.0	17.0	6.0	16.00	24.07%	3.85
11	100.0	2.0	6.0	6.40	14.71%	0.94
12	100.0	32.0	6.0	17.40	13.29%	2.31
13	100.0	17.0	2.6	13.00	24.07%	3.13
14	100.0	17.0	9.4	16.20	14.07%	2.28
15 (C)	100.0	17.0	6.0	14.00	18.79%	2.63
16 (C)	100.0	17.0	6.0	15.00	19.64%	2.95
17 (C)	100.0	17.0	6.0	14.60	18.14%	2.65
18 (C)	100.0	17.0	6.0	15.60	17.71%	2.76

**Table 4 foods-13-03646-t004:** Media and fermentation conditions tested in 250 mL shake flask and 10 L STBR by wild-type strain.

Media Composition	250 mL Shake Flask	10 L STBR
SCM: Sugarcane molasses (g/L)	66	90
CSL: Corn steep liquor (g/L)	24.5	25
RYC: Residual yeast cream (g/L)	6	15
SS: Sea salts (g/L)	10	10
MSG: Monosodium glutamate (g/L)	-	2
Fermentation conditions		
Operation mode	Batch	Batch
Working volume	50 ml	7 L
Headspace volume	200 ml	3 L
Agitation speed (rpm)	120	400–800
Impeller type	-	Rushton
Aeration rate (vvm)	-	0.5–1.0
Dissolved oxygen (DO)	-	30%
Temperature (°C)	28	25–28
pH of liquid media	6	5.5–6.0
Inoculum rate (%)	10	10
Cultivation time (h)	120	120

**Table 5 foods-13-03646-t005:** Kinetics in batch cultivation for lipid-rich biomass production by *S. limacium* SR21 in shake flasks and bioreactor using agro-industrial by-products.

Kinetic Parameters	250 mL Shake Flask	10 L STBR
*μ*max (day^−1^)	0.59	0.66
*τd* (h)	28.41	25.11
*rx* max (g/L.h)	0.23	0.64
*rx* total (g/L.h)	0.12	0.28
*rx* overall (g/h)	0.01	1.94
*rs* max (g/L.h)	0.51	0.74
*rs* total (g/L.h)	0.33	0.43
*rs* overall (g/h)	0.02	3.03
*rp* max (g/L.h)	0.09	0.35
*rp* total (g/L.h)	0.04	0.12
*rp* overall (g/h)	0.002	0.87
*Q_S_* max (h^−1^)	0.12	0.09
*Q_P_* max (h^−1^)	0.01	0.02
*Xi*: Initial biomass (g/L)	2.36	5.98
*Xf*: Final biomass (g/L)	17.30	39.29
Δ*X*: Biomass (g/L)	14.94	33.31
*Si*: Initial substrate (g/L)	43.27	60.25
*Sf*: Final substrate (g/L)	3.65	8.25
Δ*S*: Residual substrate (g/L)	39.62	52.00
*Pi*: Initial product (g/L)	0.08	0.10
*Pf*: Final product (g/L)	4.41	14.98
Δ*P*: Product (g/L)	4.33	14.88
Y_X/S_ (gbiomass/gsubstrate)	0.38	0.64
Y_P/S_ (gproduct/gsubstrate)	0.11	0.29
Y_P/X_ (gproduct/gbiomass)	0.29	0.45

## Data Availability

Data are contained within the article.

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
