# Peer review of "Sustainable Omega-3 Lipid Production from Agro-Industrial By-Products Using Thraustochytrids: Enabling Process Development, Optimization, and Scale-Up"

_foods, 2024, doi:10.3390/foods13223646_

Round 1

Reviewer 1 Report

Comments and Suggestions for Authors

This article provides a compelling overview of how Thraustochytrids, particularly Schizochytrium limacinum SR21, can be utilized for sustainable omega-3 production using agro-industrial by-products, potentially offering an economical and environmentally friendly alternative to traditional omega-3 sources. I will mention some specific minor suggestions for improving the clarity and depth of the study:

Line 13. Please write italic “Schizochytrium limacinum”.

Line 31 and Line 41: Please maintain consistency between “ω-3” and “n-3”

Line 97: Did you used a previously reported protocol?

Line 99. Please use the symbol for “⁰C”

Line 122. How was uncontrolled fermentation managed?

Line 138: Step 3 or Step III?

Line 172. Please add the name and the concentration of the food additive “by addition of food grade antifoam”

Line 213. How many replicates did you applied?

Line 357. Are these results or are they part of the method? “Table 5. Media and fermentation conditions tested in 250 mL shake flask and 10L STBR by wild-type strain.”

I recommend addressing the potential limitations of large-scale production.

Comments on the Quality of English Language

I would recommend a careful review of the English language in the text to enhance clarity and flow. Some sentences could benefit from rephrasing for smoother readability.

Author Response

Dear Reviewer 1,  
Attached, you will find a Word document containing my responses to your previous suggestions. Thank you for your valuable feedback.  

Reviewer 2 Report

Comments and Suggestions for Authors

no

Author Response

Dear Reviewer 2,  
Attached, you will find a pdf document containing my responses to your previous suggestions. Thank you for your valuable feedback.  

Reviewer 3 Report

Comments and Suggestions for Authors

Dear Authors,

I read carefully your manuscript titled “Sustainable omega-3 lipid production from agro-industrial by-products using Thraustochytrids: enabling process development, optimization, and scalable”. For scalable I believe you meant scalability.

The well-organized manuscript explains how to improve omega-3 lipid production from agro-industrial by-products using Thraustochytrids. It is ecologically friendly because it explains how to use agro-industrial waste to get useful lipids.

The Abstract summarises clearly and briefly, but very effectively, the content of the manuscript.

The Introduction begins with a general observation that explains the benefits of n-3 fatty acids. In line 31 maybe could be better to use some other word instead of renowned, for example well-known. The authors used appropriate literature but it could be refreshed with some recent references.

line 44 fish, fish oil, krill oil, and algae.

Thraustochytrids, marine protists also known as heterotrophic microalgae, have emerged as a sustainable and renewable source of high-value lipids. This is significant for the application of n-3 in different areas.

At the end of the Introduction, the authors clearly defined the aim of their study.

The methodology is very complicated demanding for authors to define the optimal quantity of agro-industrial by-products, first in 250 ml Erlenmeyer flask and then in a bioreactor. They, of course, explained the strain of Thraustochytrids they used and the mechanism of their reactivation. The bioprocess to optimize the production of biomass and lipids was explained in detail, first in the flasks and then in a stirred-tank bioreactor. The authors very successively presented the total process for biomass recovery, especially lipids.

To select the significant variables in the production of biomass and lipids the authors used the Plackett-Burman (PB) design, and at the end to optimize the production of biomass and lipids they used the central composite design.

The significant results that the authors have reached are in accordance with the methodology.

CCD allowed the generation of mathematical regression models that showed that predicted value for biomass values closely followed the trend line, while lipid values were more dispersed. Is the reason for lipids falling outside the regression area, greater variability in the lipid production data? What is the significance of that in practical terms?

The authors evaluated the desirability index and concluded that it differed for biomass and lipids content. Could you please explain?

Could you please precisely explain the sentence: the multicriteria Y1, Y2, and Y3 were ideal for considering the three response variables for biomass (g/l), lipid content (%), and lipids (g/l) with 0.87 overall desirability index?

You explained almost all the steps included in bioprocess modeling and optimization. Which step during optimization media in combination with SCM, SCL, and RYC to achieve multiple desired outcomes is the most critical step for producing biomass and lipids as well?

Is the RYC variable crucial for lipid generation? Is that the reason for RYC concentration was increased to 15 g/L during the initial hours of fermentation through various 10L bioreactors?

What is the explanation for the fact that high MSG concentrations decrease lipid content and low MSG concentrations increase DHA content after addition to the bioreactor?

Why did product-biomass yield increase from 0.29 g lipid/g biomass in a shaker to 0.45 g lipid/g biomass in a bioreactor?

The conclusion is the great part of this manuscript. The Authors succeeded in a couple of very important sentences to emphasize the importance of using agro-industrial by-products in biomass and lipids generation. The most important thing is to generate n-3 LC PUFA for use in incorporation in foods for general consumption. In this way, the Authors pointed out the significance of Thraustochytrids as an emerging biomass source for high-value n-3 PUFA.

After reading this paper I concluded that the authors made a lot of effort to indicate that the use of agro-industrial waste can help in synthesizing valuable nutrients as well as in ecological issues. For example, elongation and desaturation of n-3 LC PUFA (especially DHA) from ALA in humans is minimal.

Best regards

Comments on the Quality of English Language

The English could be improved to more clearly express the research.

Author Response

Dear Reviewer 3,  
Attached, you will find a Word document containing my responses to your previous suggestions. Thank you for your valuable feedback.  
